# The Influence of Guozhuang Dance on the Subjective Well-Being of Older Adults: The Chain Mediating Effect of Group Identity and Self-Efficacy

**DOI:** 10.3390/ijerph192114545

**Published:** 2022-11-05

**Authors:** Yuanzheng Lin, Bin Zhao, Xiujie Ma

**Affiliations:** 1School of Wushu, Chengdu Sport University, Chengdu 610041, China; 2Chinese Guoshu Academy, Chengdu Sport University, Chengdu 610041, China

**Keywords:** Guozhuang dance, subjective well-being, group identity, self-efficacy, older adults, mediating role

## Abstract

Background: In the context of the gradually accelerating aging of the population, the subjective well-being of older adults has received extensive research attention. Guozhuang Dance is a collective aerobic exercise that plays an important role in the physical activity of older Chinese adults. Studying the intrinsic relationship between Guozhuang Dance and the subjective well-being can help improve the quality of life and well-being of older adults in their later years. This study was conducted in Chengdu City, Sichuan Province, China, where many older adults practice Guozhuang Dance. Previous studies pointed out that group identity and self-efficacy can influence well-being in a collective exercise. Methods: For this study, we recruited 520 adults (male = 228, female = 292) aged 65 years or older from Chengdu who participated in Guozhuang Dance, to conduct a survey to understand the effect of this exercise on their subjective well-being. The Guozhuang Dance exercise scale, the group identity scale, the self-efficacy scale, and the subjective well-being scale were used in the study. We used SPSS for the descriptive statistical analysis, and AMOS for the structural equation modeling. Results: The results of the study show that Guozhuang Dance has a positive effect on enhancing the subjective well-being of older adults and can increase the subjective well-being through the chain mediating effect of group identity and self-efficacy. Conclusions: We suggest that effective measures should be taken to encourage older adults to participate in Guozhuang Dance, in order to enhance their subjective well-being.

## 1. Introduction

According to information released by the World Health Organization (WHO) in October 2021, the proportion of people over the age of 60 in the world’s total population will reach 22%, by 2050, and 80% of older adults will live in low- and middle-income countries [1]. According to China’s National Bureau of Statistics, the number of people aged 60 and above in China reached 267.36 million in 2021, accounting for 18.9% of the total population, among whom, 20.56 million people were aged 65 and above, accounting for 14.2% of the total population [2]. Mental diseases have become a public health concern, threatening the health of older adults in the context of the accelerating aging of the population. With increased age, psychological problems, such as anxiety [3] and depression [4] frequently occur among older adults. High levels of depression and anxiety symptoms lead to low levels of well-being [5,6]. According to the relevant research, mental illnesses in older adults can lead to a higher risk of self-harm, suicidal tendencies, cognitive impairment, Alzheimer’s, and other diseases [7,8]. This can increase their vulnerability, reduce their social activity, have a negative impact on their mental health, and decrease both their quality of life and subjective well-being [9,10].

Recently, more studies have shown that physical exercise can reduce anxiety and depression levels in older adults [11,12], promote their physical and mental health, and improve their subjective well-being [13,14]. Studying the well-being of various groups of people, including older adults, has been of great interest to researchers. For example, aerobic exercises, such as jogging [15], tai chi [16], and qigong [17] have positive effects on relieving anxiety and depression and improving the subjective well-being of older adults. Guozhuang Dance is a unique traditional ethnic Tibetan activity that is now widely popular in various cities in Western China, against a background of population flow and historical and cultural changes. The dance is especially popular in Chengdu, a city in the southwest [18]. Guozhuang Dance has been linked to improved blood circulation, increased breathing rates, and improved coordination of all body parts, according to earlier research [19], and its movements are full of festive, cheerful, and funny elements that can relieve mental stress [20]. The study found that the largest group of Guozhuang Dance practitioners is the elderly, because Guozhuang Dance not only keeps the elderly physically healthy, but most importantly, it is perfectly integrated within social culture, meaning that older adults have the chance to socialize while exercising, thus generating positive emotions, which is conducive to their subjective well-being [21].

However, there are plenty of studies on physical activity promoting subjective well-being in older adults, but there are few studies that focus on the effect of a specific sport on the well-being of older adults, such as Guozhuang Dance. In addition, there is a paucity of studies on group identity and self-efficacy promoting subjective well-being. In the context of an increasing aging population, the physical and mental health of the elderly is gradually declining. There are many older adults participating in Guozhuang Dance, so exploring the influence of Guozhuang Dance on the subjective well-being of the elderly can help them achieve the process of healthy aging and reduce the national medical burden. Therefore, through a questionnaire administered to older adults, this study aimed to investigate the effect of Guozhuang Dance on their subjective well-being and to explore whether group identity and self-efficacy mediated the relationship between them.

### 1.1. Subjective Well-Being and the Guozhuang Dance

As a pleasant emotional state, subjective well-being is usually judged by performing a comprehensive assessment of an individual’s quality of life and satisfaction [22]. Diener noted that the subjective well-being consists of three dimensions, life satisfaction, positive affect, and negative affect, and pointed out that subjective well-being is positively correlated with life satisfaction and positive affect, and negatively correlated with negative affect [23]. People who engage in long-term activities are more optimistic than those who exercise less, and they have a greater life satisfaction and happiness [24,25]. Li and Liu pointed out that exercise can directly increase individuals’ well-being, and the more exercise people do, the stronger their perception of well-being, and the better their life satisfaction [26].

Guozhuang Dance, as a mass activity, has an important role in promoting the cardiovascular and physical and mental health of older adults [27]. The main feature of Guozhuang Dance is self-entertainment; the crowd engages in dancing and singing, the scene is cheerful and harmonious, which is achieved through the “strength”, “speed”, and “range” embodied in the dancers’ artistic feelings, giving them a sense of self-satisfaction, as well as a relaxed and happy mood [20]. The dance is popular among older adults because it is simple and easy to learn; has no constraints, in terms of region, location, and the number of individuals; it also has cultural and other qualities, such as variety, intermingling, integration of production and life, fitness, and entertainment [28]. Regular exercise with Guozhuang Dance can have a good effect on people’s physical health, improve their mental outlook, cultivate their aesthetic taste, relieve fatigue, and keep them happy [19]. Liu et al. pointed out that Guozhuang Dance exercise allows people to express emotions, release negative emotions, meet their psychological needs, generate spiritual resonance, and gain self-confidence, and that it has a huge promoting effect on people’s physical and mental health [29].

To sum up, due to its attribute of being physically active, Guozhuang Dance has the same value and function as other physical fitness activities, and the view that physical exercise is positively correlated with the subjective well-being has been confirmed by many studies. However, due to the huge differences in the forms, styles, and characteristics of different physical activities, they have different effects on the subjective well-being.

Thus, hypothesis H1 is proposed: Guozhuang Dance has a positive impact on the subjective well-being of older adults.

### 1.2. Mediation of the Group Identity

Guozhuang Dance exercise in groups may increase the subjective well-being of older adults. According to Tajfel’s theory, social identity is an important factor that distinguishes between inner and outer groups [30]. When individuals realize that they are in a group that shares their own behaviors and goals, they devote their own emotions and meaning to the group, and have a sense of identity and belonging. The stronger an individual’s sense of identity in a group, the easier it is to engage in group activities [31]. Yang believes that collective dance provides a communal living space for dancers in which they actively interact and communicate, generating shared emotions and values, and thus intensifying the sense of group identity [32]. Thus, collective dance helps to create possibilities for the construction of a new publicness and identity, which has a positive impact on the self-satisfaction and identity of the elderly [33]. Further research shows that there is a positive correlation between group identity and subjective well-being [34], and individuals with high levels of group identity are more likely to experience well-being [35]. A sense of group identity can satisfy individuals’ psychological needs within the group. The activation of an in-group identity provides individuals with psychological resources, thus enhancing their well-being to a certain extent [36].

Thus, research hypothesis H2 is proposed: Group identity plays a mediating role between Guozhuang Dance and the subjective well-being of older adults.

### 1.3. Mediation of Self-Efficacy

Self-efficacy refers to the intrinsic motivation to participate in physical activities and is an important factor that promotes behavioral changes. According to the social cognitive theory proposed by Bandura, people’s behavior is influenced by their self-efficacy, and they are more willing to engage in activities that they think they can successfully complete [37]. Older adults with high levels of self-efficacy will be more actively engaged in physical activities [38]. At the same time, exercise can enhance self-confidence and the ability to cope with social challenges, allowing people to gain a sense of achievement in physical activities, thus enhancing their self-efficacy [39].

Self-efficacy can overcome the negative emotions and promote the subjective well-being in older adults [40,41]. At the psychological level, the emotional regulation of older adults is affected by self-efficacy, and the positive emotions can increase their mental toughness and improve their well-being [42]; at the physical level, self-efficacy can promote motivation, which contributes to the generation of subjective well-being [43]. Self-efficacy acts as an important bridge between physical activity and the subjective well-being, and a higher sense of self-efficacy promotes psychological well-being in older adults [44].

Thus, research hypothesis H3 is proposed: Self-efficacy plays a mediating role between Guozhuang Dance and the subjective well-being of older adults.

### 1.4. Chain Mediation of Group Identity and Self-Efficacy

According to a review of the relevant theories and literature, group identity and self-efficacy may be mediating variables between Guozhuang Dance and the subjective well-being of older adults. Zumeta et al. pointed out that group physical activity can increase the members’ sense of identity with the group, thereby increasing the collective efficacy [45]. In health-oriented groups, group identity affects individuals’ motivation to engage in activities and promotes an increased self-efficacy [46]. The stronger the individual identifies with a group advocating a particular behavior, the higher the level of self-efficacy, based on what the group advocates [47]. During exercise, older adults with a higher sense of identity show a higher self-efficacy, and their subjective well-being can also be enhanced [48]. This shows that group identity can affect individuals’ motivation to participate in an activity and meet certain goals, which in turn affects their subjective well-being.

Thus, research hypothesis H4 is proposed: Group identity and self-efficacy play a chain mediating role between Guozhuang Dance and the subjective well-being of older adults.

## 2. Materials and Methods

### 2.1. Hypotheses and Conceptual Model

The structural equation model for this study, based on the above literature review and the proposed hypotheses, is shown in Figure 1. The model shows the correlation between Guozhuang Dance, group identity, self-efficacy, and subjective well-being. Firstly, Guozhuang Dance is directly related to subjective well-being; secondly, Guozhuang Dance is indirectly related to subjective well-being through group identity and Guozhuang Dance is indirectly related to subjective well-being through self-efficacy; finally, Guozhuang Dance is indirectly related to subjective well-being through the chain effect of group identity and self-efficacy.

### 2.2. Participants and Procedure

In this study, a survey was used to investigate older adults who participated in Guozhuang Dance in Chengdu, Southwest China. A total of 600 questionnaires were distributed, and 520 valid questionnaires were recovered, with an effective rate of 86.67%. The demographic characteristics of the sample are shown in Table 1. Since this study explored the relationship between Guozhuang Dance and the subjective well-being of older adults, the inclusion and exclusion criteria needed to be defined. The inclusion criteria were: (1) people aged 65 years and above, (2) people participating in Guozhuang Dance exercise, (3) provision of informed consent, and (4) no communication barriers. The exclusion criteria were: (1) questionnaire completed in less than 1 min, and (2) repetition and invalid answers.

The questionnaires were distributed online and offline and took 3–5 min to fill out. Questionnaires were distributed online to designated groups through Questionnaire Star, including the instructions, and offline by two to three researchers who visited Guozhuang Dance venues. Ethical approval was obtained in advance from the Chengdu Sports University. All participants were informed of the reason for the study, how the study data would be used, and any risks associated with the study before completing the questionnaires. All questionnaires were accompanied by an informed consent form, and the subjects were asked to check “I have read the informed consent form” before completing the questionnaire. The researcher was required to explain any questions that the subjects could not understand and to provide oral interpretation in the participants’ own dialects, if necessary. Upon completion of the survey, a red packet or gift was given to the participant as a token of appreciation.

### 2.3. Control Variables

Since subjective well-being is a reflection of the overall level of quality of life of older adults and is influenced by both personal and social factors, gender, age, income, education, and marital status were used as control variables in this study to reduce the risk of statistical bias.

### 2.4. Instruments

#### 2.4.1. Guozhuang Dance Exercise Level

Referring to Liang’s Physical Activity Rating Scale (PARS-3) [49], this study measured the exercise level in three aspects: intensity, time, and frequency of Guozhuang Dancing. In this study, a 5-point Likert scale was used, with exercise intensity and frequency scored on a scale of 1–5, and exercise time on a scale of 0–4. The calculation formula of the total score is: exercise volume = intensity × time × frequency [49]. The highest score is 100 points, and the lowest score is 0 points. Physical activity was categorized as follows: high physical activity: ≥43 points; medium physical activity: 20–42 points; and low physical activity: ≤19 points. Cronbach’s alpha for this scale was 0.887.

#### 2.4.2. Group Identity

We used Cameron’s sports social identity scale, revised by Bruner and Benson [50]. The scale contains a total of nine items that evaluate in three dimensions: in-group ties, cognitive centrality, and in-group effect; an example is, “I feel like I have a strong bond with other members of the team”. The scale uses a 7-point scoring method, ranging from “strongly disagree” (1 point) to “strongly agree” (7 points), with higher scores indicating a stronger sense of identity. Cronbach’s alpha fore this scale was 0.906.

#### 2.4.3. Self-Efficacy

We adopted the Chinese version of the general self-efficacy scale, developed by Zhang and Schwarzer [51], which has good reliability and validity among older adults in China [52]. The scale contains a total of 10 items; for example, “It is easy for me to stick to my ideals and achieve my goals.” The scale has only one dimension and is scored on a 4-point Likert scale, ranging from “very incorrect” (1 point) to “very correct” (4 points), with higher scores indicating a greater self-efficacy. Cronbach’s alpha of this scale was 0.923.

#### 2.4.4. Subjective Well-Being

To more comprehensively evaluate the subjective well-being of older adults, we used the Memorial University of Newfoundland Scale of Happiness (MUNSH) [53], which has been validated among elderly Chinese adults [54,55]. The scale has a good reliability and validity and has been used in several countries to measure the mental health status of older adults. The scale consists of 24 questions, divided into four dimensions: positive affect (PA), negative affect (NA), positive experience (PE), and negative experience (NE). For each question, two points are given for “yes”, one point for “don’t know”, and no points for “no”. Item 19 includes the answers “present living place”, which is coded as two points, and “other living places”, which is zero points. Item 23 includes the answers “satisfied”, two points, and “not satisfied,” zero points. The total happiness score is calculated as PA-NA + PE-NE, with a score range of 0–48. The higher the score, the happier the person. A score of 36 or above indicates a high level of subjective well-being, a score of 12 or below indicates a low subjective well-being, and a score of 12–36 indicates an intermediate subjective well-being. Cronbach’s alpha for this scale was 0.911.

### 2.5. Statistical Analysis

We used the SPSS (23.0) and AMOS (24.0) for the data analysis. First, Harman’s one-way test was applied to determine the presence of the generalized method bias, resulting in the extraction of eight factors with eigenvalues greater than one, with the first factor explaining 34.69% (<40%) of the variance. Next, a validation factor analysis was conducted using AMOS to calculate the structural validity and the combined reliability to confirm the validity of each of the variables used in this study. Then, we analyzed the descriptive statistics and binary correlations of the variables in this study, using the t-test or the Pearson product-moment correlation coefficient. Next, the structural equation modeling was performed using the maximum likelihood estimation method of AMOS, and the fit of the model was tested. Finally, bootstrapping (*n* = 2000 bootstrap samples) was carried out to conduct a detailed analysis of the structural equation modeling to investigate the direct role between Guozhuang Dance and the subjective well-being of older adults, and the mediating roles of group identity and self-efficacy.

## 3. Results

### 3.1. Reliability and Validity Tests

Since the questionnaire of this study was developed with reference to the questionnaire of previous studies, the validation factor analysis was required to ensure that the collected data matched the hypothesized model, that the composite reliability and convergent validity of the data were good and that the factor loading values met the requirements of this study. We can see from Table 2 that the factor loadings of the four latent variables are between 0.689 and 0.868, the average variance extracted (AVE) is between 0.509 and 0.726, and the composite reliability (CR) is between 0.757–0.923. The results indicated that the measurement of the four latent variables has a good reliability and validity.

### 3.2. Descriptive Statistics and Correlations between the Main Study Variables

The t-test was performed to explore the gender differences in group identity, self-efficacy and subjective well-being. The results show that there were no differences in these variables, in terms of gender (see Table 3). In order to preliminarily explore the relationship between Guozhuang Dance exercise, group identity, self-efficacy, and subjective well-being, we conducted a correlation analysis of these variables. As the results in Table 4 show, Guozhuang Dance is significantly positively correlated with group identity, self-efficacy, and subjective well-being; group identity is significantly positively correlated with self-efficacy and subjective well-being; and self-efficacy is significantly positively correlated with subjective well-being. This result provided a basis for the subsequent mediation effect test, indicating that group identity and self-efficacy may have a mediating effect between Guozhuang Dance and subjective well-being.

### 3.3. Mediating Analysis

In order to verify the mediating role of the group identity and self-efficacy between Guozhuang Dance and subjective well-being, we conducted a fitting analysis of the conceptual framework mediation model using the AMOS 24.0 software package. The results of the model fit analysis of the mediation effect of Guozhuang Dance, group identity, self-efficacy, and subjective well-being are shown in Table 5, where GFI = 0.968, AGFI = 0.958, CFI = 0.985, NFI = 0.969, IFI = 0.985, and RMSEA = 0.032. The data all met the criteria, indicating the good fit of the model.

According to the bootstrap mediation effect test, proposed by Hayes [56], the number of repeated samplings of the original sample needs to be higher than 1000 times. If the bootstrap test shows that CI does not contain a 0 value, it means that the indirect effect is established [57]. The mediating bootstrap 95% CI effect was estimated by 2000 samplings, and the mediation effect test was carried out. The test of the mediating model (Table 6) further showed that the direct effect of Guozhuang Dance on the subjective well-being of the elderly was significant (direct effect = 0.360, 95% CI [0.254, 0.449]). The indirect effect contained three significant mediating pathways: Guozhuang Dance → group identity → subjective well-being (indirect effect = 0.129, 95% CI [0.067, 0.162]); Guozhuang Dance → self-efficacy → subjective well-being (indirect effect = 0.057, 95% CI [0.012, 0.088]); and Guozhuang Dance → group identity → self-efficacy → subjective well-being (indirect effect = 0.022, 95% CI [0.005, 0.035]).

The standardized path coefficient model of Guozhuang Dance, group identity, and self-efficacy, affecting the subjective well-being is shown in Figure 2. The path coefficient of Guozhuang Dance → subjective well-being (β = 0.36, *p* < 0.001) was significant, indicating that the dance exercise has a direct effect on the subjective well-being of older adults. The path coefficients of Guozhuang Dance → group identity (β = 0.38, *p* < 0.001) → subjective well-being (β = 0.34, *p* < 0.001) were significant, indicating that group identity has a mediating effect between the dance and subjective well-being. The path coefficients of Guozhuang Dance → self-efficacy (β = 0.41, *p* < 0.001) → subjective well-being (β = 0.14, *p* < 0.001) were significant, indicating that self-efficacy has a mediating effect between the dance and subjective well-being. The path coefficient of Guozhuang Dance → group identity → self-efficacy (β = 0.41, *p* < 0.001) → subjective well-being was significant, indicating that group identity and self-efficacy have a chain mediating effect between the dance and subjective well-being. Therefore, these results support all of the hypotheses.

## 4. Discussion

### 4.1. Direct Effect of Guozhuang Dance on the Subjective Well-Being

We found that Guozhuang Dance had a significant positive effect on the subjective well-being of older adults (β = 0.360, *p* < 0.001), supporting hypothesis 1. This shows that the higher the level of participation in Guozhuang Dance exercise, the higher the subjective well-being. This is consistent with previous research showing that Guozhuang Dance can promote physical health and improve one’s psychological state [19], and the level of exercise can significantly affect the level of well-being [26]. Previous studies have found that adults with higher levels of physical activity have a better life satisfaction and subjective well-being [58]. In addition, other studies have shown that dancing can improve bad moods and significantly reduce levels of depression in older adults [59]. In recent years, the physical and mental health of older adults has become a focus of attention around the world. As a collective exercise activity, Guozhuang Dance meets the physical and mental health needs of older adults to a large extent. First, the positive emotions brought by exercise are an important source of happiness, from the movement characteristics of Guozhuang Dance, show that everyone works in unison, in step and in high spirits during the dance, unintentionally promoting positive emotions [60]. Second, the combination of music and rhythmic exercise can promote the physical and mental health of older adults and improve their subjective well-being [61,62]. The music used during Guozhuang Dance includes allegro and adagio rhythms. The adagio rhythm is gentle and steady, and the content involves spirits, altars, living Buddhas, etc. The lyrics are fixed and short, and the overall style is more dignified, making it suitable for older adults [63]. Third, from the perspective of movement form, the dance involves a variety of hand and foot movements. During the exercise, older adults dance to the rhythm of the music, which is similar to the movement of square dancing. Square dancing has been proven to be effective in improving the well-being of participants [33], and is a very effective non-drug intervention that promotes physical and mental health and the quality of life in older adults [64]. Thus, it can be seen that, during the Guozhuang Dance exercise, older adults can feel the joy of exercise, eliminate bad emotions, and experience happiness.

### 4.2. Mediating Role of the Group Identity in the Influence of Guozhuang Dance on the Subjective Well-Being

The results show that Guozhuang Dance could positively predict the group identity (β = 0.38, *p* < 0.001), the group identity positively predicted the subjective well-being (β = 0.34, *p* < 0.001), and the group identity played a part in the mediating role in Guozhuang Dance and the subjective well-being of older adults (β = 0.129, *p* < 0.001), supporting hypothesis 2. Participation in team-based physical activity can increase the group identity of the exerciser [65], and group identity can positively predict the subjective well-being of older adults, which is consistent with previous studies [32,33]. The more happiness is enhanced through group identity, the more motivated and energetic people are to seek out more social opportunities, thereby further enhancing their well-being [66]. Therefore, in order to meet their own social and psychological needs, older adults will participate in Guozhuang Dance exercises to experience more positive emotions. Social-emotional selection theory argues that, as the remainder of life shortens, people continually refine their social networks to meet their social needs [67]. Participating in Guozhuang Dance can meet the social needs of older adults. First, it represents a strong cohesive force, emphasizing cooperation and common feelings among members of the group, focusing on harmony and the creation of a group atmosphere [60]; second, group dance helps participants increase their social intimacy in the group [68]; and finally, older adults participating in Guozhuang Dance will care for and help their peers during the exercise [60].

A sense of group identity means the members of the group are closer and more socially connected, making it easier for individuals to perceive happiness [69]. Due to the unique movement style of Guozhuang Dance, it is carried out in the form of collective exercise. People form a circle and cooperate with each other to perform consistent actions together. When participants perform the same actions, their action and perception networks are co-activated, which blurs their perception of others and self [70]. Furthermore, it requires participants to be highly social, due to the coordination of movements. They hold hands or put their hands on the shoulders of others. Through these movements, they strengthen their connections and their sense of group identity and belonging [71]. In short, since the practice of Guozhuang Dance is collective, interactions among members undoubtedly strengthen their connection, increase their intimacy, and meet their social and psychological needs. In such a form of exercise, members of the group always maintain a positive mood, and this will enhance the well-being of older adults.

### 4.3. Mediating Role of Self-Efficacy in the Effect of Guozhuang Dance on the Subjective Well-Being

The structural equation model analysis showed that Guozhuang Dance could positively predict self-efficacy (β = 0.41, *p* < 0.001), self-efficacy had a positive predictive effect on the subjective well-being of older adults (β = 0.14, *p* < 0.001), self-efficacy played a mediating role between Guozhuang Dance and the subjective well-being of older adults (β = 0.057, *p* < 0.001), supporting hypothesis 3. McAuley’s research found that exercise in older adults is positively associated with self-efficacy [72]. Parschau et al. also reported that exercise could positively predict self-efficacy [39]. In addition, the self-efficacy of older adults is positively correlated with the subjective well-being [73] and can increase the subjective well-being [42,43]. The results, regarding Guozhuang Dance in this study, are consistent with the above study and further support the previous study.

Self-efficacy refers to people’s subjective judgment regarding whether they can complete a certain activity. This study suggests that older adults with a high self-efficacy have more successful experiences, can find their own value, and have more confidence in themselves. This confidence enables them to experience more positive emotions while engaging in Guozhuang Dance. The relationship between self-efficacy and the subjective well-being depends on the interaction of cognition and emotion and is mediated by positive emotions [74]. Older adults perceive Guozhuang Dance as a healthy, pleasant, and meaningful activity. This encourages them to release their emotions during the movement, creating an emotional atmosphere. Positive emotions are uplifting and conducive to the development of self-efficacy and the subjective well-being. In addition, during the Guozhuang Dance, by measuring the intensity and effect of the exercise, older adults can make a subjective evaluation of their confidence in whether they can stick with it. This kind of efficacy evaluation plays an important mediating role between Guozhuang Dance and the subjective well-being, providing a stronger motivation for older adults to participate in the exercise.

### 4.4. Chain Mediating Role of the Group Identity and Self-Efficacy in the Effect of Guozhuang Dance on the Subjective Well-Being

Structural equation modeling revealed multiple internal associations with regard to Guozhuang Dance affecting the subjective well-being of older adults. Specifically, group identity positively predicted self-efficacy (β = 0.41, *p* < 0.001), group identity and self-efficacy played a chain mediating role between Guozhuang Dance and the subjective well-being of older adults (β = 0.022, *p* < 0.001), supporting hypothesis 4. Miller et al. believed that behavior, identity, and self-efficacy are mutually reinforcing, physical activity can prompt individuals to develop a strong sense of identity with physical activity, and identity can strengthen their goals and regulate their emotional state, thereby increasing their self-efficacy in maintaining physical activity [75]. Guan and So noted that in groups advocating specific health behaviors, individuals with a stronger sense of identity experience higher levels of self-efficacy when performing such behaviors [47]. The conclusion of the present study is consistent with the those of the above studies, further validating the link between identity and self-efficacy.

Social identity theory states that when individuals have a prominent identity in a group, they tend to behave in relation to the group. The stronger the individual’s role in the group, and the more prominent, the higher their self-efficacy [76]. As Guozhuang Dance is a healthy activity that is unanimously recognized by the group, in the process of exercise, people can perceive their own group identity and think that it is meaningful, and this enhances their motivation to exercise with the group, which ultimately increases their level of self-efficacy. In this case, group identity acts as a kind of psychological “glue”. It unites the members of the group and enables them to encourage each other, thereby transforming a sense of identity into a sense of efficacy, and this allows people to be more actively involved in the exercise of Guozhuang Dance. In addition, Bandura’s self-efficacy theory states that the effectiveness of learning and modeling a behavior depends on one’s own perceived similarity to the model [77]. The perceived similarity to others or to group members is an important concept in establishing a link between social identity and self-efficacy [47]. Therefore, this study argues that a sense of identity helps older adults feel that they are similar to others in the group, thus linking the relationships within the group to their own beliefs, prompting them to have similar motivations as other group members to perform Guozhuang Dance. We believe that Guozhuang Dance can bring more opportunities for interpersonal relationships and group living for older adults. The self-worth of older adults in the group can be improved, as well as their self-esteem, which not only enhances their sense of identity, but also strengthens their confidence in performing Guozhuang Dance. This encourages them to complete the exercise, which allows them to gain more positive emotions, thus increasing their subjective well-being and forming a virtuous circle.

In conclusion, there is a strong link between group identity and self-efficacy. Guozhuang Dance is a form of exercise that promotes the formation of closer connections among older adults in the group, and enhances their group identity. By adjusting the in-group identity, self-worth, and motivation of older adults, identity affects their self-efficacy, and ultimately their subjective well-being.

### 4.5. Contributions

This study offers four main contributions. First, this study extends the theoretical knowledge of Guozhuang Dance, in relation to the subjective well-being, by linking it to the subjective well-being in older adults. Second, this study analyzes the mechanisms underlying the influence of Guozhuang Dance on the subjective well-being and finds that group identity and self-efficacy can play a chain mediating role. Next, this study also discusses the important factors that influence the subjective well-being of older adults and provides practical references for enhancing the subjective well-being of older adults. Finally, this study also provides guidance for older adults to enhance their subjective well-being through Guozhuang Dance, so that they can apply effective methods to promote subjective well-being.

### 4.6. Limitations

This study has certain limitations. First, the group identity scale used in the study has not been used by anyone in China, and more studies should be conducted in the future to verify its feasibility in China. Second, it only discusses the mediating role of group identity and self-efficacy, and cannot fully reveal the mechanism of the effect of Guozhuang Dance on the subjective well-being of older adults. Future research can add more mediator and moderator variables to more clearly explain the impact mechanism of Guozhuang Dance on older adults, such as the moderating effect of self-esteem on identity and efficacy. Third, the representativeness and richness of the samples in this study need to be improved. Future research can add more regions and age groups to test the applicability of the results. Finally, this was a cross-sectional study, and the causal relationships between variables could not be demonstrated. Future research should conduct longitudinal intervention experiments to further investigate the causal relationship of variables.

## 5. Conclusions

The present study assessed the mediating role of group identity and self-efficacy in Guozhuang Dance on the subjective well-being of older adults. The results show that this exercise can significantly and positively predict the subjective well-being of older adults. Group identity and self-efficacy play partial mediating roles between Guozhuang Dance and the subjective well-being of older adults. Through the chain mediating effect of group identity and self-efficacy, Guozhuang Dance can have a positive relationship with the subjective well-being of older adults. These findings have both theoretical and practical implications.

Theoretically, this study enriches the research on the positive effects of Guozhuang Dance and the factors influencing subjective well-being. In particular, it reveals the mediating mechanisms underlying the relationship between Guozhuang Dance and the subjective well-being by integrating psychological and physical factors. It also further supports the idea that physical activity promotes physical and mental health in older adults. Practically, individuals should acknowledge the benefits of Guozhuang Dance and participate in exercise rationally and frequently. Considering the importance of group identity and self-efficacy, older adults should be guided to experience subjective well-being in a healthy way or through reasonable means. Finally, effective measures should be taken to encourage seniors to participate in Guozhuang Dance as a way to improve their subjective well-being.

## Figures and Tables

**Figure 1 ijerph-19-14545-f001:**
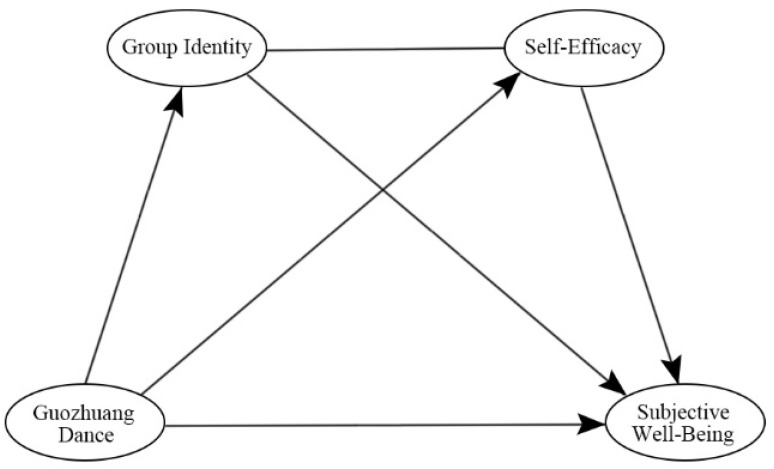
Research model.

**Figure 2 ijerph-19-14545-f002:**
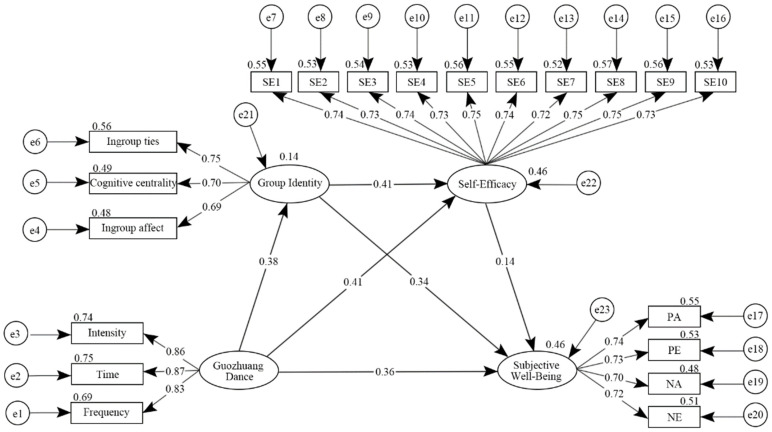
Intermediary model diagram. Note: SE, self-efficacy.

**Table 1 ijerph-19-14545-t001:** Demographic characteristics of the samples.

Variable	Frequency	Percentage (%)
Gender		
Male	228	43.85%
Female	292	56.15%
Age		
65–70	469	90.19%
70–75	38	7.31%
75–80	10	1.92%
>80	3	0.58%
Income		
<2000	203	39.04%
2000–3000	171	32.88%
3000–4000	99	19.04%
4000–5000	29	5.58%
>5000	18	3.46%
Education		
Primary school	251	48.27%
High school	196	37.69%
College and higher	73	14.04%
Marital status		
Married	484	93.08%
Unmarried	2	0.38%
Divorced	15	2.88%
Widowed	19	3.65%

**Table 2 ijerph-19-14545-t002:** Confirmatory factor analysis.

Variable	Topic	Parameter Significance Estimation				Factor Load	Reliability	CR	AVE
		Unstd.	S.E.	*t*-Value	*p*	Std.	SMC		
Guozhuang Dance	Frequency	1.000				0.829	0.688	0.888	0.726
	Time	0.982	0.032	30.229	***	0.868	0.754		
	Intensity	1.003	0.034	29.869	***	0.858	0.736		
Group identity	In-group ties	1.000			***	0.749	0.561	0.757	0.509
	Cognitive centrality	0.900	0.052	17.267	***	0.689	0.475		
	In-group effect	0.958	0.055	17.445		0.701	0.492		
Self-efficacy	Self-efficacy 1	1.000				0.740	0.739	0.923	0.545
	Self-efficacy 2	0.980	0.044	22.229	***	0.727	0.730		
	Self-efficacy 3	0.991	0.044	22.418	***	0.730	0.736		
	Self-efficacy 4	0.939	0.042	22.259	***	0.730	0.731		
	Self-efficacy 5	0.999	0.044	22.941	***	0.744	0.752		
	Self-efficacy 6	0.974	0.043	22.702	***	0.742	0.744		
	Self-efficacy 7	0.941	0.043	21.930	***	0.725	0.721		
	Self-efficacy 8	0.980	0.043	22.974	***	0.744	0.753		
	Self-efficacy 9	0.986	0.043	22.809	***	0.740	0.748		
	Self-efficacy 10	0.938	0.042	22.079	***	0.724	0.725		
Subjective well-being	PA	1.000				0.742	0.550	0.810	0.516
	NA	0.957	0.052	18.339	***	0.726	0.527		
	PE	0.943	0.049	18.296	***	0.708	0.513		
	NE	0.911	0.051	17.833	***	0.696	0.484		

Note: *** *p* < 0.001.

**Table 3 ijerph-19-14545-t003:** Results of the t-tests in different genders.

Variable		Group Identity	Self-Efficacy	Subjective Well-Being
Gender	male (*n* = 228)	4.58 ± 1.16	2.56 ± 0.71	4.03 ± 0.94
	female (*n* = 292)	4.63 ± 1.16	2.55 ± 0.72	4.12 ± 0.87
	*p*	0.563	0.949	0.130
	F	−0.579	0.064	−1.516

**Table 4 ijerph-19-14545-t004:** Descriptive statistics and correlations among the main variables.

	M	SD	Guozhuang Dance	Group Identity	Self-Efficacy	Subjective Well-Being
Guozhuang Dance	1.894	0.875	1			
Group identity	4.605	1.180	0.308 ***	1		
Self-efficacy	2.557	0.718	0.533 ***	0.470 ***	1	
Subjective well-being	4.065	0.916	0.440 ***	0.420 ***	0.447 ***	1

Note: *** *p* < 0.001.

**Table 5 ijerph-19-14545-t005:** Model fit.

	χ^2^	*df*	χ^2^/*df*	*p*	GFI	AGFI	CFI	NFI	IFI	RMSEA
Model	282.321	146	1.934	0	0.968	0.958	0.985	0.969	0.985	0.032

**Table 6 ijerph-19-14545-t006:** Mediating effect analysis.

	Standardized Value	Bootstrap LLCI	Bootstrap ULCI
Indirect effect			
Guozhuang Dance → group identity → subjective well-being	0.129	0.067	0.162
Guozhuang Dance → self-efficacy → subjective well-being	0.057	0.012	0.088
Guozhuang Dance → group identity → self-efficacy → subjective well-being	0.022	0.005	0.035
Direct effect	0.360	0.254	0.449
Total effect	0.568	0.475	0.639

## Data Availability

The data presented in this study are available upon request from the corresponding author. The data are not publicly available due to an ethical agreement with the Chengdu Sport University Social Sciences Ethics Panel, to keep them under Ma Xiujie’s personal OneDrive account, which is not accessible to the public.

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
