# Peer review of "The Influence of Guozhuang Dance on the Subjective Well-Being of Older Adults: The Chain Mediating Effect of Group Identity and Self-Efficacy"

_ijerph, 2022, doi:10.3390/ijerph192114545_

Round 1

Reviewer 1 Report

First of all, congratulations for the great contribution. This research is of great interest to society in general and to the adult population in particular.

The scientific article is developed with a high quality. Only one analysis and a different sample of the results are required. It is necessary to show the results disaggregated by gender or carry out a previous analysis to confirm that the results are not statistically different by gender.

Author Response

Dear Reviewer:

Thank you for your kind comments on our article, which were very precise and helpful for us. In accordance with your suggestions, we have made a substantial revision of this article. Below, you will find a breakdown of the responses to your comments (in italics):

Comment 1: The scientific article is developed with a high quality. Only one analysis and a different sample of the results are required. It is necessary to show the results disaggregated by gender or carry out a previous analysis to confirm that the results are not statistically different by gender.

Response: Thank you very much for your valuable suggestion, I have added tables and notes on the effect of gender on each variable in the manuscript, confirming that the statistical results are not affected by gender differences (Lines 273-275). In addition, I have made further improvements for the study design.

Thank you again for your suggestions and help, with kindest regards!

Author Response

Thank you for your kind comments on our article, which were very precise and helpful for us. In accordance with your suggestions, we have made a substantial revision of this article. Below, you will find a breakdown of the responses to your comments (in italics):

ABSTRACT:

Comment 1: Dividing in sections as described in the guidelines.

Response: We have described the summary section in segments.

INTRODUCTION:

Comment 2: Lines 41-44 need a reference, or rephrase.

Response: Thank you very much for your valuable suggestion, we have rewritten and added references in this section (Lines 45-47).

Comment 3: Lines 45-50, you’re mentioning the literature without specifying to which age band are these papers related. If it’s possible, mention the difference about the impact of the PA on young adults/adults/older adults.

Response: Thank you very much for your valuable advice, we have changed the references to show the positive impact of physical exercises on reducing anxiety and depression and enhancing the well-being of older people (Lines 48-51).

Comment 4: You have to mention some references about the impact of the dance on the well-being; then you can specify and describe the Guozhuang dance but you should not be too long. I suggest reducing the lines from 59 to 79.

Response: Thank you very much for your suggestion, we have reduced lines 59-79, and we briefly described Guozhuang Dance and added literature on the effects of Guozhuang Dance on physical and mental health and subjective well-being of older adults. (Lines 54-64). In lines 87-95 specifically describe Guozhuang Dance.

Comment 5: Please, clearly define the aims.

Response: Thank you very much for your suggestions, and we have elaborated on the research aims of this paper, including: why to study older adults, the purpose of studying older adults' well-being, and exploring the effect of mediating variables (Lines 65-75).

CHAPTER 2:

Comment 6: Line 93 needs reference.

Response: We appreciate your suggestion, we have added references (Lines 87-88).

Comment 7: This chapter should not stand by itself. In my opinion you should reduce it and make it part of the introduction. Go directly to the focus point and mention the essential things. Like this the first part of the manuscript is too long.

Response: Thank you very much for your suggestions, we have made this section part of the introduction and have reduced some of the content to focus on it.

Comment 8: The research model should be part of the methods.

Response: Thank you very much for your suggestion, we have included the study model as part of the methodology and described the model in detail (Lines 168-175).

METHODS

Comment 9: The level of the dance has been calculated through a formula. Add the reference of this formula.

Response: Thank you very much for your suggestion, we have added a reference to the formula for calculating the dance (Lines 212-213).

Comment 10: Were the MUNSH and the Cameron Scale validated and adapted for the Chinese population? If so, you should mention it also as a limitation.

Response: We appreciate your suggestions, MUNSH has been widely used in China to study subjective well-being of older adults with good reliability and validity, and I have added some literatures in the manuscript to make citations (Line 233-234). However, the Group identity scale has not been used in Chinese studies yet and is expected to be used more in future studies, so I have used it as a limitation of the manuscript (Lines 470-472).

Comment 11: Say more about the statistical analysis.

Response: Thank you very much for your suggestion, we have refined the statistical analysis and described my analysis process in more detail (Lines 247-259).

Comment 12: The methodology is poor because is lacking a control group.

Response: Thank you very much for your suggestion. According to your proposal, control groups are set up in this paper to reduce the influence of irrelevant variables on the statistical results (Lines 203-206).

RESULTS:

Comment 13: Lines 246-250 should go to the methods.

Response: Thank you for your advice, we have put this content into the method (Lines 247-250). 

Comment 14: Could you explain why you did the confirmatory analysis and why you tested the reliability and the validity of the variables and not just the correlation?

Response: Thank you very much for your question. Since our research questionnaire refers to previous research questionnaires, it is necessary to analyze the validity of the questionnaire to ensure that the data matches the model. We have supplemented the previous explanation in detail in the manuscript (Lines 262-266).

Comment 15: The model is complete but complicated; anyway, it based on your hypothesis and about some results from not validated test. I think that you should revise the entire methodology.

Response: Thank you very much for all your suggestions. As you said, our model does have some complexity, therefore, we have reduced and refined the results section and supported the hypothesis based on the results of the study model.

In addition, following your suggestions, we have improved the whole method. First, the hypothesis model was included as part of the methodology; second, we described in detail the entire process of completing the questionnaire, being the demographic characteristics of the subjects; third, we set up control groups to ensure that the study variables were not influenced by other factors; fourth, we refined the scale to ensure its applicability; and finally, we analyzed the data in a more detailed description.

Thank you again for your suggestions and help, with kindest regards!

Reviewer 3 Report

This is technically competent study involving a population of older adults. It has been designed clearly and tests hypotheses that are grounded in the extant literature. The rationale for the study (based on a demographic analysis) is established persuasively and the review of literature is appropriate.

An aspect of the manuscript that would strengthen the impact of the study would be to make a more explicit statement about the original contribution to knowledge. The findings of the study affirm what is already known about the benefits of collective physical activity. If the only element that is original is the nature of the activity (Guozhuang dance), what does contribution does the study make to advancing theoretical understanding and/or professional practice? This would not need to be a lengthy explanation but would guide the reader.

For an international audience there needs to be a clearer account of the nature of the activity. There are examples on media sites that help the uninformed reader understand better what Guozhuang dance is like. At the very least a nuanced description would add to appreciation of the physical activity.

There are other points of detail:

·         It would help to include the sex of participants in the abstract.

·         It might also help to offer an explanation of why the experiences of older adults make them different from other adults – in other words, why they are a population that needs investigating?

·         Lines 75-76 includes ‘personal values’ – to what does this refer?

·         Lines 107, 417 & 418 refer to ‘sports’. There has been some potential conceptual conflation here. It’s not clear in what way Guozhuang dance is ‘sport-like’ – and any philosophical interpretation of sport would include a necessary condition that does not seem to be met. The simplest response would be to remove this – unless there is a sporting characteristic that’s important to the discussion. But at the moment the manuscript prompts a question about this aspect.

·         Line 123 refers to a ‘specific social group’ – it might be worth explaining that this is a collection of people with a shared characteristic rather than a ‘group’ in the sense of group dynamics.

·         Line 203 – I am not sure exactly what is intended by the exclusion criteria (an example of an unrealistic answer would help).

·         Line 241 – Should the abbreviation be ‘MUNHS’?

A careful copy-edit and checking of the accuracy of the references is also necessary.

Author Response

Thank you for your kind comments on our article, which were very precise and helpful for us. In accordance with your suggestions, we have made a substantial revision of this article. Below, you will find a breakdown of the responses to your comments (in italics):

Comment 1: An aspect of the manuscript that would strengthen the impact of the study would be to make a more explicit statement about the original contribution to knowledge. The findings of the study affirm what is already known about the benefits of collective physical activity. If the only element that is original is the nature of the activity (Guozhuang dance), what does contribution does the study make to advancing theoretical understanding and/or professional practice? This would not need to be a lengthy explanation but would guide the reader.

Response: Thank you very much for your suggestions and we have added the contribution of this paper to the back of the manuscript with four points (Lines 459-468).

Comment 2: For an international audience there needs to be a clearer account of the nature of the activity. There are examples on media sites that help the uninformed reader understand better what Guozhuang dance is like. At the very least a nuanced description would add to appreciation of the physical activity.

Response: Thank you very much for your suggestions, we have described the potlatch in detail in the manuscript to improve the understanding and interest of international scholars (Lines 88-95).

Comment 3: It would help to include the sex of participants in the abstract.

Response: Thank you for your suggestion, we have described the gender of the participants in the abstract (Line 18).

Comment 4: It might also help to offer an explanation of why the experiences of older adults make them different from other adults – in other words, why they are a population that needs investigating?

Response: Thank you very much for your suggestion. In the context of population aging, the subjective well-being of the elderly is worth studying. There are a large number of elderly people participating in the Guozhuang Dance. It is meaningful to explore the happiness of the elderly through the Guozhuang Dance. Therefore, under this background, we chose the elderly as the research object (Lines 68-72).

Comment 5: Lines 75-76 includes ‘personal values’ – to what does this refer?

Response: Thanks to your question, we have rewritten personal values. Values here are the perceptions, understandings, judgments or choices that a person makes based on his or her own thinking senses (Lines 115-118).

Comment 6: Lines 107, 417 & 418 refer to ‘sports’. There has been some potential conceptual conflation here. It’s not clear in what way Guozhuang dance is ‘sport-like’ – and any philosophical interpretation of sport would include a necessary condition that does not seem to be met. The simplest response would be to remove this – unless there is a sporting characteristic that’s important to the discussion. But at the moment the manuscript prompts a question about this aspect.

Response: Thank you very much for your valuable suggestions, and we couldn't agree with you more. “Sports” in manuscripts can indeed be confusing and mislead readers. Therefore, we have removed or rewritten “sports” according to your point of view.

Comment 7: Line 123 refers to a ‘specific social group’ – it might be worth explaining that this is a collection of people with a shared characteristic rather than a ‘group’ in the sense of group dynamics.

Response: Thank you very much for your suggestion. Based on your suggestion, we replaced the term ‘specific social group’ with ‘a group similar to their own behaviors and goals’ (Lines 112-113).

Comment 8: Line 203 – I am not sure exactly what is intended by the exclusion criteria (an example of an unrealistic answer would help).

Response: Thank you for your question. We have rewritten it here, specifically ‘repetition and invalid answers’ (Line 188).

Comment 9: Line 241 – Should the abbreviation be ‘MUNHS’?

Response: Thank you very much for your suggestion, we have made changes. The full name of this scale is ‘Memorial University of Newfoundland Scale of Happiness’, hence its abbreviation ‘MUNSH’. (Line 233)

Comment 10: A careful copy-edit and checking of the accuracy of the references is also necessary.

Response: Thank you very much for your suggestions. We have carefully checked and revised the content and references of the manuscript in response to your suggestions, and confirmed the accuracy of the references.

Thank you again for your suggestions and help, with kindest regards!

Round 2

Reviewer 1 Report

The authors have modified the points raised in the previous review

Author Response

Dear Reviewer 1:

Thank you very much for your detailed reviews of our article, your suggestions are very helpful to improve our article. We have made minor revisions to the article.

Thank you again from the bottom of my heart for your careful comments on our article, with kindest regards!

Reviewer 2 Report

I thanks the authors for their deep reviewing of the manuscript and I ask them to use more data into the discussion to support their affirmations.

Author Response

Dear Reviewer 2:

First of all, thank you very much for your affirmation of our first round of revisions, your comments were very helpful to us.

Second, thank you for your careful review of our article in the second round, and we have revised it according to the changes you gave us. We have added data from the study results in the Discussion section to better support our hypothesis (Lines 319-310, Lines 348-351, Lines 380-384, Lines 407-410). We have also carefully examined the discussion section and made minor changes to make the content more specific, clear, and logical.

Finally, once again, thank you from the bottom of my heart for your continued meticulous reviews of our article, which has been greatly enhanced with your help, with kindest regards!